# Lipotype acquisition during neural development is not recapitulated in stem cell–derived neurons

Anusha B Gopalan[1,2], Lisa van Uden[1], Richard R Sprenger[3], Nadine Fernandez-Novel Marx[4], Helle Bogetofte[3,5], Pierre A Neveu[1], Morten Meyer[5,6,7], Kyung-Min Noh[4], Alba Diz-Muñoz[1], Christer S Ejsing[1,3]

During development, different tissues acquire distinct lipotypes that are coupled to tissue function and homeostasis. In the brain, where complex membrane trafficking systems are required for neural function, specific glycerophospholipids, sphingolipids, and cholesterol are highly abundant, and defective lipid metabolism is associated with abnormal neural development and neurodegenerative disease. Notably, the production of specific lipotypes requires appropriate programming of the underlying lipid metabolic machinery during development, but when and how this occurs is unclear. To address this, we used high-resolution MS[ALL] lipidomics to generate an extensive time-resolved resource of mouse brain development covering early embryonic and postnatal stages. This revealed a distinct bifurcation in the establishment of the neural lipotype, whereby the canonical lipid biomarkers 22:6-glycerophospholipids and 18:0-sphingolipids begin to be produced in utero, whereas cholesterol attains its characteristic high levels after birth. Using the resource as a reference, we next examined to which extent this can be recapitulated by commonly used protocols for in vitro neuronal differentiation of stem cells. Here, we found that the programming of the lipid metabolic machinery is incomplete and that stem cell–derived cells can only partially acquire a neural lipotype when the cell culture media is supplemented with brain-specific lipid precursors. Altogether, our work provides an extensive lipidomic resource for early mouse brain development and highlights a potential caveat when using stem cell–derived neuronal progenitors for mechanistic studies of lipid biochemistry, membrane biology and biophysics, which nonetheless can be mitigated by further optimizing in vitro differentiation protocols.

## Introduction

Lipids are a diverse category of biomolecules that constitute and functionalize membranes, serve as energy reservoirs, and function as signaling molecules. Although different tissues and cell types produce and maintain characteristic lipid compositions, termed lipotypes (Hicks et al, 2006; Harayama et al, 2014; Capolupo et al, 2022), how they are acquired during the course of development is largely unknown. Furthermore, the functional implications of maintaining a distinct lipotype are not understood, although the existence of cell type– and tissue-specific lipid homeostasis across organisms points to conserved mechanisms governing critical functions (Yamashita et al, 2014; Bozek et al, 2015).

Lipotype acquisition and homeostasis are of particular interest in neuroscience because of the enrichment of distinct lipid molecules in the brain, which is functionally coupled to the sophisticated membrane trafficking system of neural cell types (Davletov & Montecucco, 2010; Puchkov & Haucke, 2013; Lauwers et al, 2016; Ingólfsson et al, 2017). Studies have shown severe impairments of neuronal development in mice lacking the lipid binding protein SCAP (Verheijen et al, 2009), as well as the lipid metabolic enzymes CerS2 (Imgrund et al, 2009) and DAGL (Gao et al, 2010), which are part of the cholesterol biosynthesis, ceramide (Cer) biosynthesis, and diacylglycerol breakdown pathways, respectively. Defects in sphingolipid metabolism have also been associated with neural tube defects in embryos (Stevens & Tang, 1997; Missmer et al, 2006), the degeneration of motor neurons (Bejaoui et al, 2001; Dawkins et al, 2001), and myelination defects (Fewou et al, 2005). Moreover, the loss of ether-linked phosphatidylethanolamines (PE O-, i.e., plasmalogens) and sulfatides has been linked to age-related neurodegeneration in diseases such as Alzheimer's disease (Han et al, 2001, 2002; Tu et al, 2017). Biochemical studies have also shed light on the various roles of sphingomyelin (SM) and the ganglioside GM1 (Tettamanti et al, 1996; Fan et al, 2021), and potential lipid

[1]Cell Biology and Biophysics Unit, European Molecular Biology Laboratory, Heidelberg, Germany   [2]Faculty of Biosciences, Candidate for Joint PhD Degree Between EMBL and Heidelberg University, Heidelberg, Germany   [3]Department of Biochemistry and Molecular Biology, Villum Center for Bioanalytical Sciences, University of Southern Denmark, Odense, Denmark   [4]Genome Biology Unit, European Molecular Biology Laboratory, Heidelberg, Germany   [5]Department of Neurobiology Research, Institute of Molecular Medicine, University of Southern Denmark, Odense, Denmark   [6]Department of Neurology, Odense University Hospital, Odense, Denmark   [7]BRIDGE, Department of Clinical Research, University of Southern Denmark, Odense, Denmark

Correspondence: diz@embl.de; cse@bmb.sdu.dk

signaling molecules such as retinoids, terpenoids, steroids, and eicosanoids, in triggering differentiation programs in neural stem cells (Bieberich, 2012). Furthermore, studies performed in vitro (Cao et al, 2009; Pinot et al, 2014) and in vivo (Janssen et al, 2015) have demonstrated that docosahexaenoic acid (DHA, 22:6), found in membrane glycerophospholipids (GPL), is important for neurogenesis and the formation of synapses (Salem et al, 2001; Innis, 2007). Thus, lipids have been identified as key players across neurological development, physiology, and disease.

Decades of research on brain lipids have established that the mammalian brain has a unique lipotype characterized by high levels of 22:6-containing GPL, 18:0-containing sphingolipids, and cholesterol (O'Brien et al, 1964; O'Brien & Sampson, 1965a; Fitzner et al, 2020). These canonical lipid biomarkers are conserved between mammalian species, ranging from rodents to humans (Bozek et al, 2015), and present in all neural tissues and most of the cell types (O'Brien et al, 1964; O'Brien & Sampson, 1965a, 1965b; Fitzner et al, 2020) (Fig S1A–J). In particular, these three categories of lipid biomarkers are enriched in neuron-rich gray matter (O'Brien et al, 1964; O'Brien & Sampson, 1965a) (Fig S1A and B), as well as neurons isolated from E16.5 mouse embryos (Fitzner et al, 2020) (Fig S1E). These same markers are also prevalent in synaptic vesicles (Takamori et al, 2006) and the postsynaptic plasma membrane (Tulodziecka et al, 2016) (Fig S1H–J). Because these lipids are primarily membrane constituents, it is likely that the regulation of their levels is functionally coupled to membrane trafficking events and activities of cell surface receptors of especially mature neurons.

Although the canonical lipid biomarkers of neurons are well established, it is still largely unknown how the neuronal lipotype is acquired during development. Unraveling this can in principle come from studies using in vitro neuronal differentiation of stem cells. This, however, presents the conundrum that fully mature neurons in vivo are not able to de novo synthetize the canonical 22:6ω3-containing lipids themselves (Moore et al, 1991; Kim, 2007) and that cell culture media used for in vitro neuronal differentiation are devoid of the essential fatty acid 22:6ω3 and are instead supplemented with the fatty acid 18:3ω3 (Bardy et al, 2015). Moreover, stem cell–derived neurons also do not down-regulate the expression of ceramide synthases 5 and 6, which produce 16:0-containing sphingolipids instead of the neuron-specific 18:0-containing sphingolipids (Fig S10). Thus, albeit stem cell–derived neurons acquire unique morphological hallmarks and express specific protein markers, it is at the present unclear whether the lipotype of stem cell–derived neurons in vitro is comparable to that of mature neurons and brain tissues.

To shed light on how the neural lipotype is acquired during development, and to assess the extent to which this can be recapitulated during in vitro neuronal differentiation, we generated a detailed lipidomic resource for early brain development in mice, starting from the embryonic stage of E10.5, where the brain tissue first becomes accessible for dissection, and up to the postnatal stage P21. This revealed that the high levels of canonical neural lipid biomarkers 22:6-GPL and 18:0-sphingolipids begin to be established already at the embryonic stage in utero, coinciding with extensive neurogenesis (E10.5–E15.5) (Caviness, 1982; Finlay & Darlington, 1995), whereas the increase in cholesterol occurs

postnatally. Using the resource as a reference, we examined to which extent this can be recapitulated by commonly used protocols for in vitro neuronal differentiation of mouse embryonic stem cells (mESCs) and human induced pluripotent stem cells (hiPSCs). Here, we found that stem cell–derived neurons could only recapitulate a partial neuronal lipotype, and only upon supplementing the culture medium with brain-specific lipid metabolic precursors. Taken together, our resource shows that early mouse brain development coincides with extensive lipidome remodeling, starting already at the embryonic stage, and that in vitro neuronal differentiation using standard protocols yields immature neurons that are only partially committed at the lipid metabolic level.

## Results

### Lipotype acquisition during early mouse brain development

To identify lipid hallmarks associated with neural development, we compiled a comprehensive lipidomic resource of mouse brain development. To this end, we microdissected the whole mouse brain region at the developmental stage E10.5, as well as excised cerebral hemispheres from E15.5, P2, and P21 mice (Fig 1A). These biopsies were analyzed by high-resolution MS$^{ALL}$ lipidomics (Almeida et al, 2015; Sprenger et al, 2021). Overall, this analysis identified and quantified 1,488 lipid molecules encompassing 26 lipid classes (Supplemental Data 1). Statistical analysis revealed that 451 distinct lipid molecules in the cerebral hemispheres (30% of detected lipids) were significantly changed in abundance across the developmental timeline with a twofold or greater difference between E10.5 and P21 (ANOVA followed by multiple hypothesis correction using the Benjamini–Hochberg procedure, $P \le p_c = 0.04$) (Fig 1B), of which 133 (9%) and 318 (21%) lipids increased and decreased, respectively. Principal component analysis showed a clustering of samples according to the time point and tissue, with only a weak distinction between the E10.5 brain tissue and the rest of the embryo (Figs 1C and S2). Similarly, the distinction between cerebral hemispheres and the rest of the brain tissues was not pronounced at E15.5 but clearly increased over the course of development as each brain region acquired a more specialized lipotype.

Assessing the bulk abundance of lipid classes in the brain tissue across development, we observed a 1.5-fold increase in cholesterol and a twofold increase in PE O-, as well as a corresponding decrease in phosphatidylcholines (PC) in the postnatal phase between P2 and P21. Low levels of cholesterol esters (CE) and triacylglycerol present at E10.5 were further diminished thereafter. The total level of phosphatidylinositols (PI), precursors of signaling phosphoinositides, gradually decreased in abundance throughout development (Fig 1D).

To systematically examine whether the lipids increasing in abundance have common molecular traits, we carried out Lipid feature ENrichment Analysis (LENA) (Sprenger et al, 2021), akin to gene ontology analysis of gene transcripts and proteins. This demonstrated that membrane GPL ($P = 8 \times 10^{-10}$) and sphingolipids (SP, $P = 7 \times 10^{-3}$) are enriched among the increasing lipid molecules,

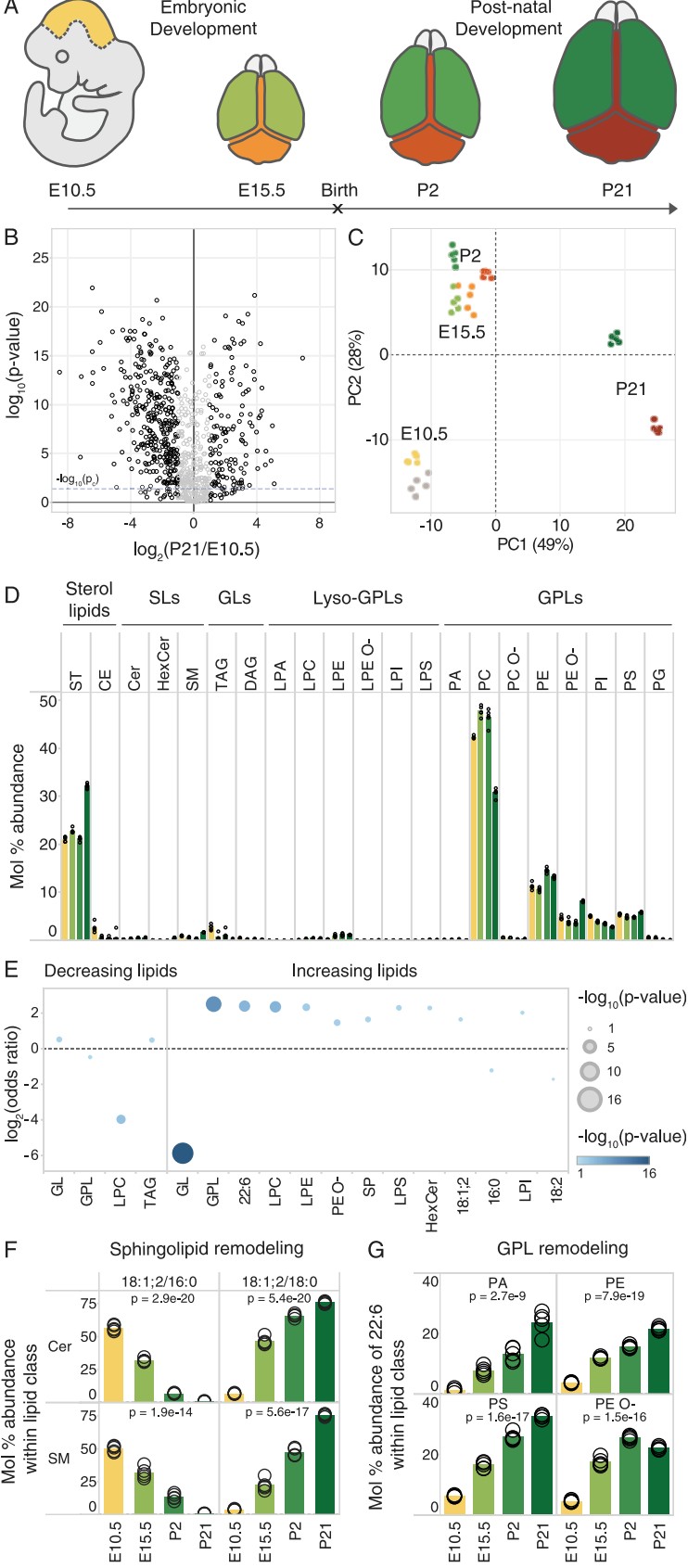

**Figure 1. Lipotype acquisition during brain development.**
**(A)** Mouse brain tissue was collected at the indicated time points and analyzed by in-depth $MS^{ALL}$ lipidomics. The colors denote distinct brain regions that were sampled at each developmental stage. **(B)** Volcano plot of molecular lipid species. The fold change of each lipid within its lipid class is calculated between the E10.5 whole brain and the P21 cerebral hemisphere. **(C)** Principal component analysis of lipid abundance across samples, colored as per the schematic in (A). **(D)** Profile (mol%) of different lipid classes across developmental time. Bars representing the mean value for each sample group are colored in accordance with panel (A). **(E)** Lipid feature ENrichment Analysis of features among decreasing and increasing lipids, where $-\log_2$ (odds ratio) is plotted in the y-axis, and the color and size correspond to the $-\log_{10}$ ($P$-value). **(F)** Profile of the most abundant Cer and SM species across samples, containing an 18:1;2 sphingoid chain and either a 16:0 or an 18:0 acyl chain. **(G)** Abundance of 22:6 (DHA) among fatty acyls constituting phosphatidic acid (PA), PE, PE O-, and PS species. Data were collected on n = 5 replicates. $P$-values were obtained using a one-way ANOVA test, and changes were considered significant if $P \leq p_c = 0.04$, as determined by the Benjamini–Hochberg procedure.

whereas storage glycerolipids (GL, $P = 1 \times 10^{-16}$) are depleted from this pool. Moreover, we found that the lysolipids LPC, LPE, LPS, and LPI are enriched in the pool of increasing lipids, in addition to hexosylceramides (HexCer) and PE O- lipids. The polyunsaturated structural attribute 22:6 ($P = 6 \times 10^{-6}$), corresponding to DHA, is significantly enriched among the increasing lipids, as are sphingolipids with a sphingosine 18:1;2 chain ($P = 2 \times 10^{-2}$) (Fig 1E). Inspecting the molecular timelines of the canonical brain lipid biomarkers showed that the sphingolipids Cer 18:1;2/18:0 and SM 18:1;2/18:0, as well as 22:6-containing phosphatidylserine (PS), PE, and PE O- species, already reach their expected high levels within the early developmental period studied here (Fig 1F and G).

For all time points, we also subjected the remainder of the brain (after removal of the cerebral hemispheres) to lipidomic analysis and found similar lipidomic changes as in the cerebral hemisphere (Fig S3A–E). In addition, the rest of the E10.5 embryos after removal of the brain region were also analyzed (Fig S4A–C), revealing a lipotype similar to the brain region at this time point (with only 13 of 1,219 [1%] analyzed lipids showing a significant twofold or larger change in abundance between the E10.5 brain and the remaining tissue). This suggests that the neural lipotype acquisition has not yet begun as of E10.5, making it a suitable starting point for profiling the lipidomic changes that concur with neural development.

Taken together, our lipidomic resource provides data on the molar abundance of individual lipid species with annotation of individual fatty acyl chains, thereby providing a comprehensive compendium of over a 1,000 molecular lipid species. We observed that neural development coincides with a stepwise lipotype acquisition. Specifically, our results reveal that 18:0-sphingolipids and 22:6-GPL (i.e., PS, PE, and PE O-) already become enriched in utero, which coincides with the onset of neurogenesis (Caviness, 1982; Finlay & Darlington, 1995), whereas enrichment of cholesterol occurs postnatally. The comparison with published data from primary neurons and glial cells further demonstrates that the lipid biomarkers observed early in development by us are present in most of the cell types isolated from the brain and are strongly enriched in neurons (Fig S1E and F).

## Lipidomic characterization of neuronal differentiation in vitro

Given that stem cell–derived neurons are commonly used for studying neuronal fate determination and acquisition of specific functions, we next examined to what extent in vitro neuronal differentiation yields neurons with lipidomic hallmarks akin to the neural lipotype observed in vivo. To this end, we made use of a commonly used protocol (Bibel et al, 2007) where mESCs are grown as suspended aggregates and differentiated into neuronal progenitors (i.e., immature neurons) over a time period of 12 d (Fig 2A) (Tiwari et al, 2012, Gehre et al, 2020; Song et al, 2021; Muckenhuber et al, 2023). The rationale for not going beyond 12 d is that longer culture time concurs with a loss in overall cell viability. Notably, in our hands, 84% ± 1% cells were positive for β-tub III, indicating the high purity of the neuronal progenitors on day 12 (Fig 2B).

Along this timeline, we analyzed cells by $MS^{ALL}$ lipidomics at 0, 4, 8, and 12 d after the onset of differentiation. Overall, the lipidomic analysis afforded quantitative monitoring of 1,355 lipid molecules encompassing 26 lipid classes (Supplemental Data 1). Of these, the levels of 431 lipids (32%) were significantly altered over the course of differentiation, with a twofold or greater difference between day 0 and day 12 (ANOVA followed by a Benjamini–Hochberg procedure, $P$-value ≤ $p_c$ = 0.023), with 200 (15%) and 231 (17%) increasing and decreasing, respectively (Fig 2C).

At the bulk lipid class level, we observed a significant increase in the storage lipids CE and triacylglycerol on day 4 and day 8, indicating that the cells were storing fat at this stage (Fig 2D). LENA demonstrated that the structural features GPL, 16:1, PE O-, and PS were enriched in the pool of increasing lipids ($P = 8 \times 10^{-6}, 1 \times 10^{-3}, 1 \times 10^{-2}, 1 \times 10^{-2}$). The increase in 16:1 is indicative of de novo lipid synthesis (Lin & Smith, 1978; Chakravarty et al, 2004; Freyre et al, 2019), which coincides with the removal of serum from the culture medium and a switch to using glucose as the primary substrate for de novo lipogenesis on day 8 (Supplemental Data 1). Furthermore, the polyunsaturated features 20:4 and 22:4 were depleted among the pool of increasing lipids ($P = 4 \times 10^{-3}, 3 \times 10^{-2}$) (Fig 2E).

Lastly, we inspected the temporal dynamics of the canonical neural lipid biomarkers during the in vitro neuronal differentiation. We found that only cholesterol is significantly increased, from 19 mol% on day 0 to 22 mol% on day 12 (Fig 2D), reaching similar values as in the developing embryonic brain (21 mol% at E10.5 and 23 mol% at E15.5), but far from those observed in neural tissue (32 mol% at P21). Notably, all 22:6-GPL are relatively low in abundance at all time points and most of these are also progressively reduced during the in vitro differentiation (Fig 2G). A similar trend was observed for the sphingolipids Cer 18:1;2/18:0 and SM 18:1;2/18:0, which was offset by an increase in Cer 18:1;2/16:0 and SM 18:1;2/16:0 (Fig 2F). Notably, 16:0-sphingolipids were found to decrease considerably during brain development in vivo, while being replaced by 18:0-sphingolipids (Fig 1F). Apart from the lack of lipid remodeling, we also observed that the molecular composition of PIs in the cells differed from that of the brain tissue, with elevated levels of 18:1 chains and reduced levels of 20:4 chains in the PI molecules (Fig S5).

In summary, our analysis shows that although the differentiated mouse stem cells resemble neurons by morphology and express specific protein markers on day 12 of differentiation, they do not acquire a unique lipotype akin to the brain tissue or primary neurons (Figs 1 and S1). In fact, the lipidome of cells at day 12 is most similar to that of the E10.5 brain (Fig S6), where the neural lipotype acquisition has not yet begun (Fig S4).

## Lack of canonical lipid markers is a general feature of in vitro neuronal differentiation

Prompted by our finding that neuronal progenitors generated by culturing mESCs in embryoid bodies (Bibel et al, 2007) do not acquire a neural lipotype, we examined lipotype acquisition in two other lineages of stem cell–derived neurons: (1) in vitro differentiation of mESCs cultured in an adherent monolayer and stimulated with retinoic acid (Ying et al, 2003) (Fig S7A) and (2) differentiation of human iPSCs to a population of predominantly dopaminergic neurons (Bogetofte et al, 2019), because the canonical brain lipid biomarkers are known to be conserved among mammals, ranging from mice to humans (Fig S1). Briefly, the human iPSCs are differentiated into neural stem cells (NSCs) through a neural rosette-

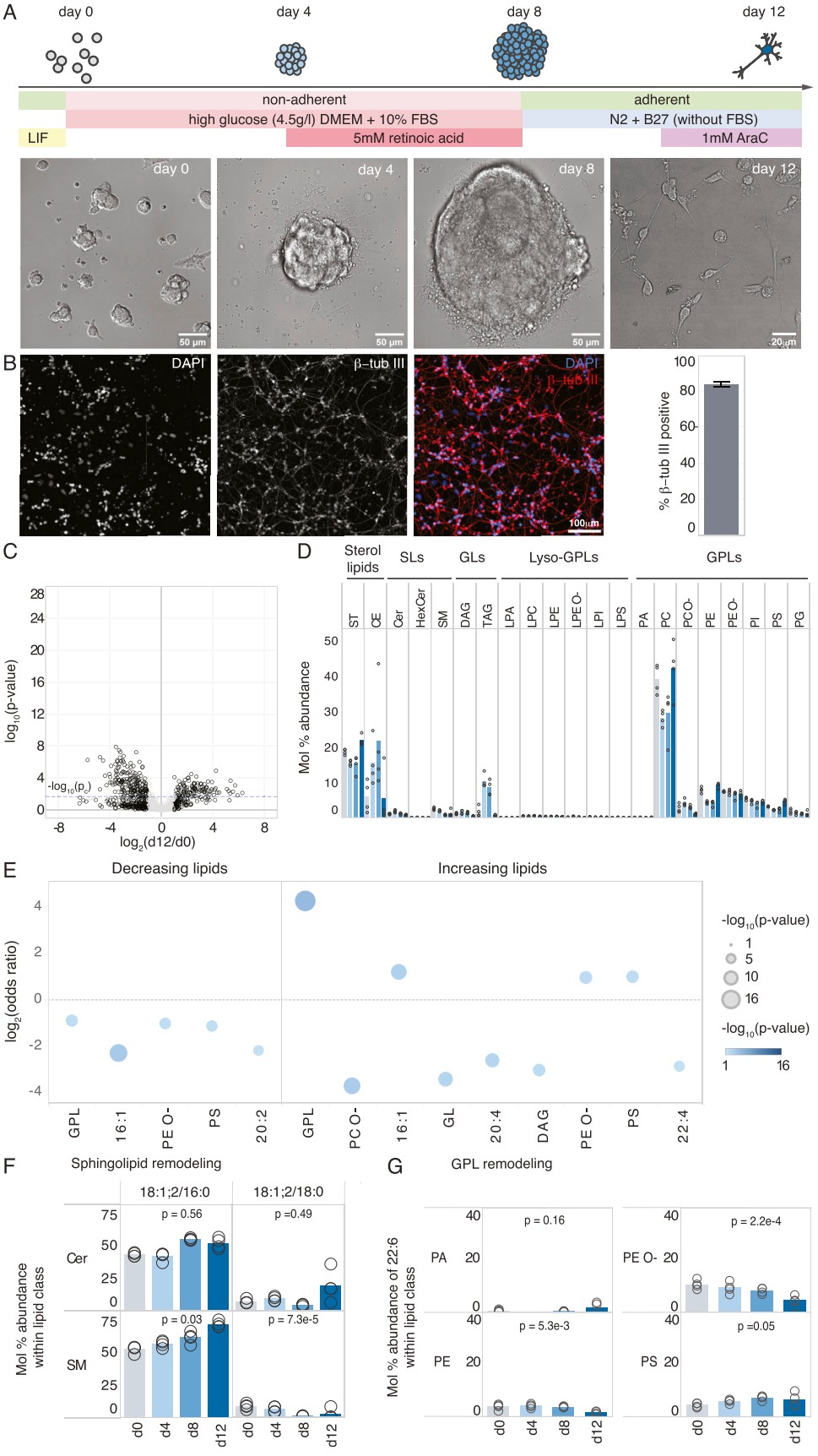

**Figure 2. Lack of lipotype acquisition during neuronal differentiation in vitro.**
**(A)** Schematic of the neuronal differentiation protocol for mouse embryonic stem cells with representative bright-field images at stages in the differentiation protocol where samples were collected for lipidomics.
**(B)** Representative fluorescence images of cells on day 12 stained for DAPI and the neuronal marker βTub-III are shown. Quantification of βTub-III–positive cells is shown on the right, based on 16 images from n = 2 biological replicates. **(C)** Volcano plot of molecular lipid species. The fold change in mol% of each molecular lipid species is calculated between day 0 and day 12 of differentiation. **(D)** Profile of different lipid classes across time. **(E)** Lipid feature ENrichment Analysis of the pool of decreasing and increasing lipids. **(F)** Profile of sphingolipids of interest within their respective lipid classes. **(G)** Profile of 22:6 (DHA) among fatty acyls constituting PA, PE, PE O-, and PS species. Data were collected on n = 4 replicates. *P*-values were obtained using a one-way ANOVA test, and changes were considered significant if $P \leq p_c = 0.023$ as determined by the Benjamini–Hochberg procedure.

based protocol (Swistowski et al, 2009). The NSCs are then further differentiated for up to 25 d to postmitotic neurons with Sonic hedgehog (Shh) stimulation to induce dopaminergic specification (Bogetofte et al, 2019).

The neuronal lineage of these cells was confirmed by immunostaining of β-tub III and Microtubule Associated Protein 2 (MAP2) on day 25, where 86.4% ± 1.0% and 67.9% ± 0.9% of the cells were positive for β-tub III and MAP2, respectively (Fig S8A and B). In both cases, the differentiated cells failed to acquire the characteristic lipotype of neural tissue (Figs S7B–D and S8C–E), despite presenting morphological features of neurons, expressing neuron-specific mRNA and protein biomarkers (e.g., β-tub III, MAP2, Sox1, NeuN), and having synaptic activity (Bibel et al, 2007; Bogetofte et al, 2019). These findings suggest a general failure of in vitro neuronal differentiation models in prompting cells to acquire the characteristic lipotype of the brain tissue and especially neurons in vivo. This discrepancy highlights a challenge in using in vitro differentiated neuronal progenitors for mechanistic studies of lipid metabolic programming and lipotype acquisition. Moreover, it prompts the need to develop new in vitro differentiation protocols that can adequately commit stem cells to acquire a neural lipotype. We note here that the standard culture media for in vitro neuronal differentiation of mESCs, containing the B27 supplement, are devoid of the essential polyunsaturated fatty acids DHA (22:6ω3) and arachidonic acid (20:4ω6). Instead, the B27 supplement contains their respective precursors, linolenic acid (18:3ω3) and linoleic acid (18:2ω6) (Brewer & Cotman, 1989; Brewer et al, 1993). The lack of 22:6ω3 is particularly surprising given that 22:6ω3 deficiency in mothers during pregnancy and in children is known to negatively impact growth and cognitive development (Lauritzen et al, 2016).

### Fatty acid supplementation partially establishes a neural lipotype in vitro

Studies using primary rat neurons and astrocytes have shown that the polyunsaturated fatty acids 22:6ω3 and 20:4ω6, found to be abundant in neural lipids, are produced by astrocytes from the precursors, 18:3ω3 and 18:2ω6, and are thereafter salvaged by neurons (Moore et al, 1991; Kim, 2007). This indicates that the external supply of fatty acids to in vitro differentiated stem cells is important for lipid metabolic remodeling and accretion of the neural lipotype. Thus, we attempted to recapitulate the lipotype acquisition observed in vivo by supplementing the culture media of differentiating mESCs with relevant neural lipid precursors.

Specifically, we supplemented the cells with a mix of 20 $\mu M$ DHA (for the production of 22:6-PE, PE O-, and PS lipids), 10 $\mu M$ arachidonic acid (for the biosynthesis of 20:4-PI lipids), and 30 $\mu M$ stearic acid (for the production of 18:0-sphingolipids). We added the fatty acid mixture to the cell culture media from day 8 of differentiation (Fig 3A). We then collected the fatty acid–supplemented cells, as well as untreated control cells on day 12 for lipidomics.

The lipidomic analysis identified and quantified 1,200 lipid molecules from 26 lipid classes (Supplemental Data 1). We found that among the 310 lipid molecules (26%) significantly altered in abundance by twofold or more (a $t$ test followed by the Benjamini–Hochberg procedure, $P$-value ≤ $p_c$ = 0.029), 124 (10%)

were increased and 186 (16%) were reduced. 112 (90%) of the increased lipids likely featured a polyunsaturated chain (using the criteria that at least one of the acyl chains had ≥4 double bonds or that the total number of double bonds in the lipid ≥4 for lipids whose acyl chain composition could not be determined) (Fig 3B). The incorporation of 22:6 into PE, PE O-, and PS closely resembled that seen in the P21 brain, with the molar abundance of 22:6 reaching 20 mol% in PE, 25 mol% in PE O-, and 31 mol% in PS (as compared to 22, 22, and 33 mol% in the P21 cerebral hemisphere) (Fig 3D). The incorporation of 20:4 into PI also resembled that seen in the brain, with the molar abundance of 20:4 reaching 38 mol% (as compared to 43 mol% in the P21 cerebral hemisphere) (Fig S9B). The stearic acid supplementation promoted only a modest increase in 18:0-containing sphingolipid species, reaching 28 mol% for ceramides and 12 mol% for SMs (as compared to 76 mol% for both ceramides and SMs in the P21 cerebral hemisphere) (Fig 3C). We note that fatty acid supplementation did not alter the bulk level of individual lipid classes (Fig S9A).

In summary, it appears that the lipid metabolic machinery underpinning the neural lipotype is partially established in the in vitro differentiated neuronal progenitors. On the one hand, it is correctly programmed to be able to take up and incorporate the essential polyunsaturated 22:6 and 20:4 into membrane GPL. On the other hand, the metabolic branch responsible for sphingolipid production appears to require more rewiring as it fails to produce the high levels of 18:0-containing sphingolipids seen in the developed brain tissue. Together, our results pinpoint two key factors responsible for lipotype acquisition, namely, cell-intrinsic enzymatic activities of the underlying lipid metabolic machinery and cell-extrinsic lipid building blocks such as polyunsaturated fatty acids (i.e., derived from the cell culture medium in vitro and the neighboring cell types in tissues in vivo).

## Discussion

In this study, we generated a comprehensive lipidomic resource of mouse brain development, starting at the early embryonic stage E10.5. This resource compliments previous lipidomic investigations of brain regions, cell types, and membrane fractions obtained from postnatal pups and adolescent animals (Breckenridge et al, 1972; Dawson, 2015; Lauwers et al, 2016; Tulodziecka et al, 2016; Fitzner et al, 2020), and adds to the emerging multi-omics picture of neural development (Yousefi et al, 2021). Our time series analysis allowed us to identify neural lipid biomarkers that increase in abundance throughout development, starting in utero (22:6-glycerophospholipids and 18:0-sphingolipids), whereas cholesterol increases postnatally (Fig 1). The specific lipotype acquisition that we observe in the developing mouse brain underscores the important role of lipids and their metabolism in the acquisition of neural functions in vivo. Moreover, the early onset of lipotype acquisition evident at E15.5 indicates that this process is closely coupled to cell differentiation in the brain.

Using the lipidomic resource as a reference, we examined to what extent commonly used protocols for in vitro neuronal differentiation of mESCs and human iPSCs concur with extensive lipid metabolic remodeling and acquisition of a neural lipotype. Despite

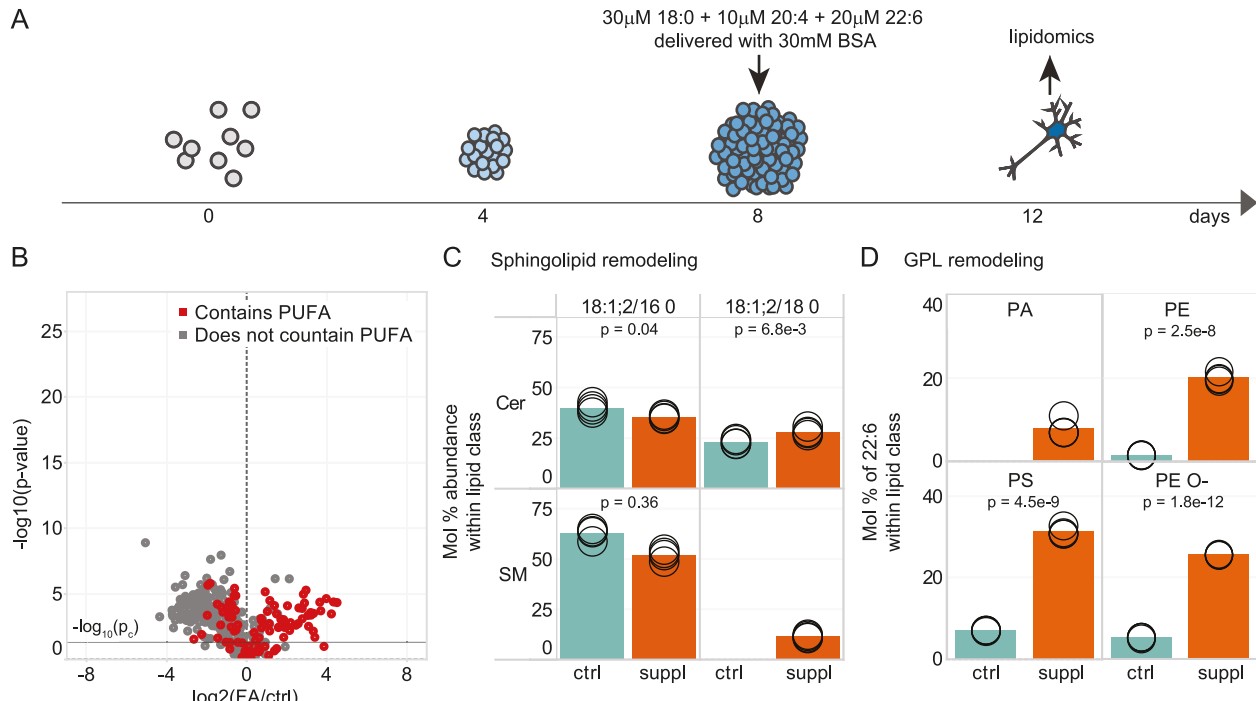

**Figure 3. Incorporation of fatty acids into the cell's lipotype.**
**(A)** Schematic of the differentiation protocol where cells were supplemented with fatty acids. **(B)** Volcano plot of molecular lipid species. The fold change in mol% abundance of each molecular lipid species was calculated between the FA-supplemented and control cells. Red circles correspond to lipid species containing four or more double bonds. **(C)** Profile of Cer and SM 18:1;2/16:0 and 18:1;2/18:0 in control (cyan bar) and FA-supplemented (orange bar) cells on day 12 of differentiation. **(D)** Profile of 22:6 among fatty acyls constituting PA, PE, PE O-, and PS species. Data were collected on n = 4 replicates. P-values were obtained using a t test and considered significant if $P \leq p_c = 0.029$, as determined by the Benjamini–Hochberg procedure.

the evident acquisition of neuronal morphology and the expression of neuronal markers in vitro (Figs 2, S7, and S8) (Izant & McIntosh, 1980; Haendel et al, 1996; Katsetos et al, 2003; Daubner et al, 2011), we did not find lipidomic changes similar to those seen across brain development. Two factors could contribute to this incomplete lipotype acquisition, namely, insufficient programming of the underlying lipid metabolic machinery required for the biosynthesis of neural lipids, the absence of required metabolic precursors, or a combination thereof. It has previously been shown that DHA (22:6) is produced in astrocytes and is thereafter salvaged by neurons (Moore et al, 1991; Kim, 2007). Our fatty acid supplementation confirms that the lack of 22:6-GPL is indeed due to a deficiency in the culture medium, as supplementation with 20 µM DHA allows cells to produce 22:6-GPL to a level comparable with the neural lipotype (Fig 3). On the contrary, supplementing the cells with stearic acid (18:0) does not result in high levels of 18:0-sphingolipids. It is known that the Cer synthase CerS1 is specific for stearoyl-CoA (Venkataraman et al, 2002), which results in the production of 18:0-sphingolipids, whereas the synthases CerS5 and CerS6 are responsible for 16:0-sphingolipid production. During brain development, one observes a 35-fold increase in the expression of Cers1 and a down-regulation of Cers5 and Cers6 compared with embryonic tissue (Sladitschek & Neveu, 2019) (Fig S10A). In contrast, during in vitro neuronal differentiation, between days 8 and 12, Cers1 expression increases only by fivefold, and contrary to expectation, Cers6 expression is up-regulated and Cers5 expression is

unchanged (Gehre et al, 2020) (Fig S10B). This could underpin the observation that 16:0-sphingolipids remain elevated, whereas brain-specific 18:0-sphingolipids only increase marginally, despite supplementation with stearic acid. Overall, this suggests that appropriate programming of the sphingolipid metabolic machinery is not fully established in stem cell–derived neuronal progenitors.

Previous work has indicated that homeoviscous adaptation can lead to an increase in cholesterol levels to preserve membrane packing when cells of various cell types, including neurons, are supplemented with high levels of PUFAs (Sinensky, 1974; Ernst et al, 2016; Leventhal et al, 2020). This finding is of particular interest in the context of the brain and neurons, as they have relatively high levels of cholesterol and PUFAs (Figs 1 and S3). However, our data show that cholesterol increases only postnatally and not concomitantly with the increase in PUFAs seen throughout development. Moreover, fatty acid supplementation did not lead to an increase in cholesterol in the differentiating mESCs (Fig S9A), in line with some other cell types (Zech et al, 2009).

Although the cholesterol level increases in some of the standard protocols used in this study (Figs 2D and S7B), the final levels are far from those observed in vivo. Previous work has shown that astrocytes are also producers of cholesterol and neurons are recipients (Pfrieger & Ungerer, 2011; Valenza et al, 2015; Ferris et al, 2017), again highlighting the importance of lipid exchange between cell types in the brain.

In summary, here we have outlined the lipidomic landscape of brain development in mice by tracing the abundance of more than 1,000 lipids over developmental time. This has allowed us to identify two key changes that begin in the embryonic phase of development, namely, an increase in 22:6-GPL and the replacement of 16:0-sphingolipids by 18:0-sphingolipids. In contrast, we find that the cholesterol level in the brain increases only postnatally. Such orchestrated lipidome remodeling suggests that the differentiation and maturation of brain cells in vivo are coupled to lipid metabolic changes. Although primary neurons from E16.5 embryos show these lipid hallmarks (Fitzner et al, 2020), our attempt to recapitulate this in vitro using neuronal differentiation models to study the molecular mechanisms responsible for lipid metabolic commitment has demonstrated that the acquisition of neural lipid hallmarks is lacking in neuronal progenitors generated in vitro. To address this deficit, we added metabolic precursors for producing the canonical neural lipids, but this only partially established the neural lipotype. Thus, future work is needed to systematically improve in vitro neuronal differentiation protocols, and perhaps consider the need for co-cultures, if they are to be used for mechanistic investigations of lipid biochemistry, membrane biology and biophysics. Our approach provides an important framework for further optimization of differentiation protocols to make in vitro neural cells that more closely recapitulate their in vivo counterparts.

## Materials and Methods

### In vivo study in mice

Mice were housed in the Laboratory Animal Resources Facility at EMBL Heidelberg in accordance with the guidelines of the European Commission, revised Directive 2010/63/EU and AVMA Guidelines 2007, under veterinarian supervision. No procedure was performed on live animals. The mothers carrying E10.5 and E15.5 embryos, as well as P2 and P21 pups, were euthanized following a protocol approved by the EMBL Institutional Animal Care and Use Committee.

### Chemicals and lipid standards

Chloroform, methanol, and 2-propanol (Rathburn Chemicals), and ammonium formate (Fluka Analytical) were all of HPLC-grade. Lipid standards were purchased from Avanti Polar Lipids and Larodan Fine Chemicals.

### Neuronal differentiation of mESCs in embryoid bodies in vitro

mESCs (129XC57BL/6J generated from male 129-B13 agouti mice) were initially cultured on a feeder layer of mouse fibroblast cells from CD1 mice in ESC media containing Knockout DMEM (Cat. #10829018; Gibco) with 15% EmbryoMax FBS (Cat. #ES009-M; Merck) and 20 ng/ml leukemia inhibitory factor (LIF from EMBL Protein Expression and Purification Core Facility), 1% non-essential amino acids (Cat. #11140050; Gibco), 1% GlutaMAX (Cat. #35050061; Gibco), 1 mM sodium pyruvate (Cat. #11360070; Gibco), 1% (50 U/ml) Pen/

Strep (Cat. #15070063; Gibco), and 143 $\mu$M $\beta$-mercaptoethanol (Cat. #21985023; Gibco). They were cultured over three passages after which the feeder cells were selectively depleted from culture by allowing them to adhere to tissue culture dishes for 10 min. The unadhered mESCs were replated for further propagation where they were differentiated into neuronal progenitor cells over the course of 12 d according to the protocol in Bibel et al (2007) (Fig 2A). For this, the mESCs were grown in suspension in non-adherent dishes containing differentiation media with high glucose DMEM (Cat. #11965092; Gibco), 10% FBS (Cat. #26140079; Gibco), and no LIF, but otherwise identical in composition to the ES medium. This results in the formation of embryoid bodies that grow larger over the course of 8 d. On day 4, 5 $\mu$M retinoic acid was added to induce neuronal differentiation. On day 8, embryoid bodies were dissociated with trypsin (Cat. #25300054; Gibco) and cells were plated at a density of 140,000 cells/cm$^2$ on plates precoated with poly-D-lysine and laminin-511 (Cat. #LN511; BioLamina) in N2B27 media (high glucose DMEM supplemented with 1% N2 [Cat. #17502048; Gibco], 1% B27 [without vitamin A, Cat. #17504044; Gibco], 1% [50 U/ml] Pen/ Strep, and 1 mM sodium pyruvate [Cat. #11360070; Gibco]). On day 10, cells were treated with 1 $\mu$M cytosine arabinoside (AraC, Cat. #C1768; Merck) to kill proliferating, non-differentiated cells.

Media were changed every day before day 0 because of rapid cell proliferation in the stem cell state and every 2 d thereafter, with special care to prevent dissociation of the embryoid bodies because of shear and care to avoid exposing the plated neuronal progenitors to air. Cells were placed at 37°C with 5% CO$_2$ under all medium conditions.

### Fatty acid supplementation

Stock solutions of fatty acids to be supplemented were prepared in ethanol (40 mM 18:0, 32.8 mM 20:4, and 60 mM 22:6) and pipetted dropwise into N2 media containing 3 mM fatty acid–free BSA (Cat. #A8806; Merck) at 37°C with constant stirring to obtain final concentrations of 3 mM 18:0, 1 mM 20:4, and 2 mM 22:6. This 100x stock of the FA-supplemented N2 media was aliquoted and stored at −20°C. On day 8 of the differentiation protocol, it was added to the N2 culture medium in a 1:100 ratio.

### Neuronal differentiation of mESCs in a monolayer in vitro

mESCs were maintained as undifferentiated stem cells in an adherent monolayer in N2B27 media (a 1:1 mixture of DMEM/F12 [without Hepes and with glutamine, Cat. #131331028; Gibco] and Neurobasal medium [Cat. #21103049; Gibco] supplemented with 0.5x N2 [17502001; Gibco], 0.5x B27 [without vitamin A], 0.25 mM L-glutamine [Cat. #25030149; Gibco], 0.1 mM $\beta$-mercaptoethanol [Cat. #21985023; Gibco], 10 mg/ml BSA fraction V [Cat. # 10735078001; Merck], 10 mg/ml Human recombinant Insulin [Cat. #91077C; Merck], and 1% Pen/Strep) +2iLIF (10 ng/ml leukemia inhibitory factor + 3 $\mu$M CHIR99021 + 1 $\mu$M PD0325901 from Tocris, Cat. # 4423 and 4192, respectively). The cells were grown on 0.1% gelatin-coated dishes and usually seeded at a density of 0.5–1.5 × 10$^4$/cm$^2$. The medium was replaced every day, and the cells were trypsinized and reseeded every 2 d in their undifferentiated state. For monolayer differentiation, the cells were similarly seeded and kept in N2B27

without the 2iLIF for 24 h, after which the media were supplemented with 1 $\mu$M retinoic acid and refreshed every 24 h.

## Neuronal differentiation of human iPSCs in vitro

The human iPSC line XCL-1 (XCell Science Inc.) was differentiated into NSCs by XCell Science Inc. using a 14-d protocol where iPSCs were initially differentiated in suspension as embryoid bodies followed by the formation of attached neural rosettes. These NSCs were then isolated and expanded (Swistowski et al, 2009). The NSCs were propagated using Geltrex (Cat. #A1413202; Thermo Fisher Scientific)-coated plates in Neurobasal medium (Cat. #21103049; Thermo Fisher Scientific) supplemented with NEAA (Cat. #11140050; Thermo Fisher Scientific), GlutaMAX-I (Cat. #35050038; Thermo Fisher Scientific), B27 supplement (Cat. #17504044; Thermo Fisher Scientific), penicillin–streptomycin (Cat. #15140; Thermo Fisher Scientific), and bFGF (Cat. #233-FB; R&D Systems). Cells were enzymatically passaged with Accutase (Cat. #A1110501; Thermo Fisher Scientific) when they were 80–90% confluent. Neuronal differentiation was achieved by culturing NSCs in DOPA Induction and Maturation Medium (Cat. #D1-011; XCell Science) according to the manufacturer's instructions by supplementing with 200 ng/ml human recombinant Sonic hedgehog (Cat. #100-45; PeproTech) from days 0–10 and passaging cells at days 0, 5, and 10 onto poly-L-ornithine (Cat. #3655; Sigma-Aldrich)– and laminin (Cat. #23017015; Thermo Fisher Scientific)-coated plates at a density of 50,000 cells/cm$^2$.

## Immunostaining

mESCs differentiated into neurons for 12 d were fixed in 4% (vol/vol) paraformaldehyde (Cat. #28908; Thermo Fisher Scientific) for 20 min. Excess paraformaldehyde was quenched with 30 mM glycine for 5 min, and coverslips were washed three times with PBS. Cells were permeabilized with 0.1% Triton X-100 (Cat. # 3051.4; Carl Roth) and blocked with 0.5% BSA (Cat. # 8076.4; Carl Roth) for 30 min at RT. Cells were incubated with a primary antibody against $\beta$-tubulin III (Cat. # ab78078; Abcam) diluted in 0.5% BSA 1:200 and incubated for 1 h at RT. A secondary antibody conjugated to an Alexa Fluor 594 dye (Cat. # A-11005; Invitrogen) was used for detection. Cells were stained with DAPI 5 $\mu$g/ml (Cat. # D9542; Sigma-Aldrich) for 5 min, and coverslips were mounted onto glass slides with ProLong Gold (Cat. # P36934; Invitrogen). Images were acquired with a Nikon Eclipse Ti fluorescence microscope using the 20x objective.

Human iPSC-derived neurons, on differentiation day 25 from NSCs, were fixed for 15 min at RT in 4% (wt/vol) paraformaldehyde (Cat. #158127; Sigma-Aldrich) and permeabilized with 0.1% Triton X-100 (Cat. #9002-93-1; Sigma-Aldrich). Unspecific binding was blocked with 10% goat serum (Cat. #S26; Millipore), and cultures were incubated overnight at 4°C with primary antibodies diluted in TBS/10% goat serum: mouse anti-$\beta$-tubulin-III (#T8660; Sigma-Aldrich) or mouse anti-MAP2 (Cat. # M1406; Sigma-Aldrich). Incubation with secondary antibodies Alexa Fluor 555 goat anti-mouse IgG (Cat. #A21422; Molecular Probes) was done at 1:500 in TBS/10% goat serum for 2 h at RT. Cell nuclei were counterstained with 10 $\mu$M 4″, DAPI dihydrochloride (Cat. #D9542; Sigma-Aldrich). Coverslips were mounted onto glass slides with Pro-Long Diamond Mounting Medium (Cat. #P3690; Molecular Probes).

To estimate the proportion of differentiated neurons, the ratio of neuronal marker–positive cells to the total number of cells (DAPI) in each field of view was determined and averaged over several images.

## Sample collection for lipidomics

Brain tissue samples for lipidomics were collected from WT CD-1 mice from the EMBL breeding colonies at four different time points in development, namely, the embryonic stages E10.5 and E15.5 and the postnatal stages P2 and P21 (Fig 1A). No procedure was performed on live animals. The mothers carrying E10.5 and E15.5 embryos, and P2 and P21 pups were euthanized by IACUC-approved methods, after which tissues were collected for lipidomics.

At E10.5, each brain sample consisted of a pooled brain region from five embryos to obtain enough samples for lipidomics. At all remaining time points, the brain from a single mouse embryo or pup was dissected to obtain two samples: one with cerebral hemispheres and another with the rest of the brain. All tissues were washed with PBS, flash-frozen in liquid nitrogen, and stored at –80°C until lipid extraction.

Mouse cells were collected for lipidomics on days 0, 4, 8, and 12 of differentiation in embryoid bodies of mESCs. On day 0, cells were collected after trypsinization. On days 4, 8, and 12, suspended embryoid bodies from one 6-cm dish each were collected by a cell scraper. In the case of monolayer differentiation of mESCs, cells in one well of a six-well plate were collected per sample. In each case, the cells were pelleted, washed twice in Hepes–KOH buffer, suspended in 200–300 $\mu$l 155 mM ammonium formate, flash-frozen in liquid nitrogen, and stored at –80°C until lipid extraction.

Human iPSC-derived neurons were collected after treatment with Accutase on differentiation days 0, 10, and 25 (from the NSC stage). Cells were collected in ice-cold PBS, pelleted, washed with 155 mM ammonium acetate (Cat. #A7330; Sigma-Aldrich), flash-frozen in liquid nitrogen, and stored at –80°C until lipid extraction.

## Lipid extraction

Lipids were extracted from the samples as described previously (Almeida et al, 2015). In short, all samples were thawed at 4°C and homogenized by sonication (and mechanically using an ultra-turrax in addition to this, in the case of brain samples). Samples were spiked with an internal standard (IS) mix prepared in-house and containing known amounts of synthetic lipid standard (Table S1). Lipid extraction was performed by two-step extraction (Sampaio et al, 2011), first by partitioning the sample between 155 mM aqueous ammonium formate and chloroform/methanol (10:1, vol/vol) and then using the aqueous fraction to partition against chloroform/methanol (2:1, vol/vol). The solutions were shaken using a ThermoMixer (Eppendorf) set at 1,400 rpm and 4°C during both steps (for 2 and 1.5 h, respectively) and their organic phases collected. Solvents were then removed by vacuum evaporation, leaving deposits of lipids extracted in the 10:1- and 2:1-extracts, respectively. The 10:1- and 2:1-extracts were dissolved in chloroform/methanol (1:2, vol/vol).

## Shotgun lipidomics

MS[ALL] lipidomic analysis was performed as previously described (Almeida et al, 2015). In short, mass spectra of the lipid extracts were recorded in both positive and negative ion modes using an Orbitrap Fusion Tribrid mass spectrometer (Thermo Fisher Scientific) equipped with a robotic nanoflow ion source, TriVersa NanoMate (Advion Biosciences). Aliquots of 10:1-extracts were diluted with 2-propanol to yield an infusate composed of chloroform/methanol/2-propanol (1:2:4, vol/vol/vol) and 7.5 mM ammonium formate for positive ion mode analysis. The 10:1-extracts were also diluted with 2-propanol to yield an infusate composed of chloroform/methanol/2-propanol (1:2:4, vol/vol/vol) and 0.75 mM ammonium formate for negative ion mode analysis. Aliquots of the 2:1-extracts were diluted with methanol to yield an infusate composed of chloroform/methanol (1:5, vol/vol) and 0.005% methylamine for analysis in the negative ion mode. High-resolution mass spectra of intact ions (MS[1]) were recorded using an Orbitrap mass analyzer, precursor ions from each 1-Da window within the detected range were fragmented, and the mass spectra of the fragments (MS[2]) were recorded using the Orbitrap mass analyzer. Lipid molecules and fragment ions were identified using ALEX[123] (Pauling et al, 2017; Ellis et al, 2018). The molar abundance of lipid molecules was quantified by normalizing their MS[1] intensities to that of spiked-in internal lipid standards and by scaling the known concentration of the internal standard (Table S1). Lipid quantification and downstream analysis of fatty acid composition and LENA were carried out on the SAS9.4 platform (SAS) as described previously (Sprenger et al, 2021).

## Statistical analysis

In the analysis of time-course data, replicates from identical conditions were considered to constitute a sample group (resulting in nine sample groups among the mouse brain samples and four sample groups among cell culture samples). ANOVA testing with these sample groups was performed to identify significant changes in the mol% abundance of lipids. An unpaired $t$ test was used to test for significant differences in the lipidome of day 12 neurons between control and fatty acid–supplemented conditions.

To minimize false positives because of multiple hypothesis testing, we calculated a Benjamini–Hochberg critical value $q$ for every $P$-value with a false-positive rate of 0.05, and changes were considered significant for all $P \leq p_c$, where $p_c$ is the largest $P$-value that is smaller than its corresponding $q$-value. The statistical tests were performed using R, and the principal component analysis was done using ClustVis (Metsalu and Vilo, 2015). Tableau Desktop (Tableau Software) was used for data visualization.

## Data Availability

RNA-seq data from Gehre et al (2020) and Sladitschek and Neveu (2019) were accessed on the ArrayExpress repositories with accession numbers E-MTAB-6821 and E-MTAB-4904, respectively. The lipidomic dataset generated in this work can be found as Supplemental Data 1, and analysis codes are available on request.

## Supplementary Information

## Acknowledgements

We thank Martin Hermansson for useful discussions on lipidomics, Emilia Esposito, Juan Carlos Boffi, and Vikram Singh Ratnu for providing us with dissected mouse brain tissues, and Eva Pillai and Martin Bergert for critical reading of the article. This research was supported by the Danish Council for Independent Research | Natural Sciences (DFF—6108-00493) and the VILLUM Center for Bioanalytical Sciences (VKR023179) to CS Ejsing, and the European Molecular Biology Laboratory (EMBL) to K-M Noh, CS Ejsing, and A Diz-Muñoz. H Bogetofte and M Meyer were supported by the Innovation Fund Denmark (BrainStem: 4108-00008A), the Danish Parkinson Foundation, the Jascha Foundation (3687, 5611), the A.P. Møller Foundation for the Advancement of Medical Science (15-396, 14-427), and the Faculty of Health Sciences at the University of Southern Denmark.

### Author Contributions

AB Gopalan: formal analysis, investigation, visualization, and writing—original draft, review, and editing.
L van Uden: formal analysis, investigation, and visualization.
RR Sprenger: data curation, formal analysis, supervision, and investigation.
N Fernandez-Novel Marx: formal analysis, supervision, and investigation.
H Bogetofte: formal analysis and investigation.
PA Neveu: resources, supervision, funding acquisition, methodology, and writing—review and editing.
M Meyer: resources, supervision, funding acquisition, methodology, and writing—original draft.
K-M Noh: resources, supervision, funding acquisition, and methodology.
A Diz-Muñoz: conceptualization, resources, supervision, funding acquisition, methodology, project administration, and writing—original draft, review, and editing.
CS Ejsing: conceptualization, resources, supervision, funding acquisition, investigation, project administration, and writing—original draft, review, and editing.

### Conflict of Interest Statement

The authors declare that they have no conflict of interest.

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
