## [Reviewer comments · Life Science Alliance]

Life Science Alliance

Lipotype acquisition during neural development is not recapitulated in stem cell-derived neurons

Anusha Gopalan, Lisa van Uden, Richard Sprenger, Nadine Fernandez-Novel Marx, Helle Bogetofte, Pierre Neveu, Morten Meyer, Kyung-Min Noh, Alba Diz-Munoz, and Christer Ejsing

DOI: <https://doi.org/10.26508/lsa.202402622>

Corresponding author(s): Christer Ejsing, European Molecular Biology Laboratory and Alba Diz-Munoz, EMBL

Review Timeline:	Submission Date:	2024-01-26
	Editorial Decision:	2024-01-29
	Revision Received:	2024-02-07
	Accepted:	2024-02-08

Transaction Report:

Please note that the manuscript was reviewed at *Review Commons* and these reports were taken into account in the decision-making process at *Life Science Alliance*.

Reviews

Review #1

Gopalan et al use quantitative, comprehensive lipid mass spectrometry of mouse brain tissue isolated at various time points in embryonic and postnatal development. They then go on to use the same quantitative analysis of mouse and human stem cells differentiated in vitro into neurons to define the lipid composition of these cultures.

****Major Comments:****

1. As mentioned above, it is difficult to assess whether the discrepancy in the lipotype acquisition between in vivo mouse brain development and stem cell differentiation is due to metabolic differences in the in vitro differentiation as the authors state or is due to a lack of the stem cells to actually acquire a neuronal phenotype. Perhaps showing more clearly that the protocols for neuronal differentiation work efficiently and/or how they compare to brains dissected would be helpful in stating that the lipotype is different. The protocol referenced here (Bogetofte et al) only gives ~30% TH+ positive DA neurons in their manuscript. What cell type the other 70% of the cells are is something that could be discussed as means of "diluting" out the lipotype seen in these cultures. Perhaps the 30% TH+ DA neurons do attain the "correct" lipotype, but the lipidomic analysis can not detect this due to the contaminating effects of the non-differentiated cells. In this work, it would be nice to see what percentage of the cells differentiate into the expected cell type rather than referencing previous manuscripts. As differentiation protocols and originating cell sources are highly variable and error-prone, it's difficult to know what the lipotype results are actually reporting on.

Furthermore, discussion about these differentiation techniques and how well they represent functional neurons is warranted. The papers referenced here don't show 100% differentiation into the phenotypes that are described in this work such that the lipotype finding is not the only suggestion of a "general failure of in vitro neuronal differentiation models". Maybe a discussion of how the lack of ability to attain the neuronal lipotype due to the metabolic deficiencies discussed here could be causative to the inability to full recapitulate the neuronal phenotype is useful for the reader.

2. From the discussion and work here is unclear why the stearate feeding of the stem cells did not result in an increase in the 18:0-containing sphingolipids. The authors state that the appropriate metabolic pathways are not fully established and go on to look at the CerS expression levels across the differentiation timeline. It appears that the results presented in Fig S7 counter the authors' interpretation of the lipotype and more discussion here would be nice to clarify this discrepancy.

****Minor comments:****

1. I find the data presentation of the LENA analysis to be difficult to follow (Fig 1E). In my opinion, the p-value is not the most important bit of information in this graph, though having it on the y-axis with other pertinent information encoded by colors or arrows being disguised. I would rather see the data on the x-axis that is above a certain p-value (denoted in the figure legend) plotted with the direction and magnitude of change shown.

2. In the PCA in Fig 1, what are the loadings that define the variable PC1 and PC2? What is predominantly changing the P21 samples that lead to such a large shift if most of the data shown in the subsequent panels are not changing much between P2 and P21.

This work provides a nice reference for the complex lipidomes in embryonic and postnatal murine brain development. The details of the lipotype changes during development are well laid out and will of no doubt be of great use across a variety of scientific fields. While I found the in vivo data to be compelling, interesting, and useful, the lack of controls for the in vitro stem cell differentiation work makes this particular data set and comparison less useful. Further work to identify the limitations of the stem cell differentiation protocols as a valid comparison to in vivo brain development need to be done and/or the discussion of the direct comparisons between the two toned down.

Review #2

The study used a quantitative lipidomics approach which I am very familiar with. The results should be highly reproducible.

The manuscript submitted by Gopalan et al. reported a quantitative and comparative lipidomics study between

mouse brain samples from early embryonic to postnatal stages, and rodent and human stem cell-derived neurons. The authors found a couple of very unique characters only existing in brain samples, but not in stem cell-derived neurons, including 22:6-containing glycerophospholipids and 18:0-containing sphingolipids. The authors further found the brain-like lipotypes can only be partially established in stem cell-derived neurons after supplementing brain lipid precursors. These findings clearly suggest that stem cell-derived neurons might not be appropriately used to mechanistically study lipid biochemistry, membrane biology, and biophysics in brains. The study was well designed. and the manuscript was very informative and resourceful. I would suggest to accept the manuscript for publication.

1. General Statements

Stem cell-derived neurons provide unique opportunities for delineating the molecular mechanisms underlying neurodevelopment and neurological diseases as well as advancing drug development. Such research is commonly carried out using imaging, transcriptomics and other functional assays, including electrophysiology techniques that probe membrane-related properties of neurons. Despite the enormous medical relevance and potential societal impact of stem cell technologies, it is still unknown whether stem cell-derived neurons in culture are able to acquire a distinct lipotype with canonical membrane lipid biomarkers akin to brain cells *in vivo*. Moreover, it is unclear whether *in vitro* differentiation concurs with reprogramming of the lipid metabolic machinery to make stem cell-derived neurons able produce unique brain-specific membrane lipids.

To bridge this knowledge gap, we carried out a systematic and comparative lipidomics study of mouse brain development *in vivo*, covering the early embryonic stage E10.5 until the postnatal stage P21, and compared this to neuronal differentiation of stem cells *in vitro*. Surprisingly, this revealed that stem cell-derived neurons fail to acquire a distinct lipotype akin to neural cells *in vivo*. In fact, we found that this is a general problem of neuronal differentiation *in vitro* using well-established and commonly used protocols for mouse embryonic stem cells (mESCs) as well as human induced pluripotent stem cells (hiPSCs). To investigate this apparent deficit, we probed the lipid metabolic landscape by supplementing cells with fatty acid precursors of canonical brain lipids. This revealed that a combination of cell-intrinsic (metabolic enzymes) and cell-extrinsic factors (metabolic precursors obtained from the environment) are behind the inability of stem cell-derived neurons to acquire a brain-like lipotype. Specifically, we find that the *in vitro* differentiation protocols do not adequately reprogram sphingolipid metabolism, which is known to be of paramount importance for neural function and when defective prompts some of the most devastating neurological diseases.

We thank the Reviewers for their useful feedback on our manuscript. We have addressed the Reviewers' comments and revised our manuscript accordingly. A point-by-point response is provided below.

Reviewer comments:

Reviewer #1 (Evidence, reproducibility and clarity (Required)):

Gopalan *et al* use quantitative, comprehensive lipid mass spectrometry of mouse brain tissue isolated at various time points in embryonic and postnatal development. They then go on to use

the same quantitative analysis of mouse and human stem cells differentiated in vitro into neurons to define the lipid composition of these cultures.

Major Comments:

1. As mentioned above, it is difficult to assess whether the discrepancy in the lipotype acquisition between in vivo mouse brain development and stem cell differentiation is due to metabolic differences in the in vitro differentiation as the authors state or is due to a lack of the stem cells to actually acquire a neuronal phenotype. Perhaps showing more clearly that the protocols for neuronal differentiation work efficiently and/or how they compare to brains dissected would be helpful in stating that the lipotype is different. The protocol referenced here (Bogetofte et al) only gives ~30% TH+ positive DA neurons in their manuscript. What cell type the other 70% of the cells are is something that could be discussed as means of "diluting" out the lipotype seen in these cultures. Perhaps the 30% TH+ DA neurons do attain the "correct" lipotype, but the lipidomic analysis can not detect this due to the contaminating effects of the non-differentiated cells. In this work, it would be nice to see what percentage of the cells differentiate into the expected cell type rather than referencing previous manuscripts. As differentiation protocols and originating cell sources are highly variable and error-prone, it's difficult to know what the lipotype results are actually reporting on.

Furthermore, discussion about these differentiation techniques and how well they represent functional neurons is warranted. The papers referenced here don't show 100% differentiation into the phenotypes that are described in this work such that the lipotype finding is not the only suggestion of a "general failure of in vitro neuronal differentiation models". Maybe a discussion of how the lack of ability to attain the neuronal lipotype due to the metabolic deficiencies discussed here could be causative to the inability to full recapitulate the neuronal phenotype is useful for the reader.

We thank the reviewer for this question and suggested experiments. Following the advice, we have now show immunofluorescence data of pan-neuronal markers (i.e., β -tub III or MAP2) in mESC and iPSCs. In agreement with previously published datasets from the Noh and Meyer labs (Gehre et al., 2020; Bogetofte et al., 2019), we show that the protocols we use generate a very high percentage of neurons. We have now included these images and quantifications in our manuscript as **Figs. 2B** and **S6A,B**, and include the new panels here for your reference.

Figure 2B. Representative fluorescence images of cells on day 12 stained for DAPI and the neuronal marker β Tub-III are shown. Quantification of β Tub-III positive cells is shown on the right, based on 16 images from n = 2 biological replicates.

Figure S6A,B. A) Immunofluorescence images of human iPSC-derived neurons on day 25 of differentiation from the NSC stage, labelled for the neuronal markers β -tub III and MAP2. Regions in dashed white lines are shown below. B) Percentage of cells positive for β -tub III and MAP2 as quantified from 45 images per group, from n = 3 biological replicates.

2. From the discussion and work here is unclear why the stearate feeding of the stem cells did not result in an increase in the 18:0-containing sphingolipids. The authors state that the appropriate metabolic pathways are not fully established and go on to look at the CerS expression levels across the differentiation timeline. It appears that the results presented in **Fig. S7** counter the authors' interpretation of the lipotype and more discussion here would be nice to clarify this discrepancy.

We thank the reviewer for highlighting this seemingly counterintuitive observation. We have now included a quantification of CerS mRNA from commercially available mouse tissues analysed in Sladitschek and Neveu, 2019 and compared this to the data from Gehre et al, 2020. In the mouse brain tissue, CerS1 expression is upregulated dramatically, while CerS5 and 6 are downregulated (see new panel A in **Fig. S8**). In contrast, during *in vitro* differentiation of mESCs, CerS5 is not downregulated and CerS6 is upregulated (**Fig. S8B**). Accordingly, we have expanded our discussion in the revised manuscript as follows:

“On the other hand, supplementing the cells with stearic acid (18:0) does not result in high levels of 18:0-sphingolipids. It is known that the Cer synthase CerS1 is specific for stearyl-CoA

(Venkataraman et al., 2002), which results in the production of 18:0-sphingolipids, while the synthases CerS5 and CerS6 are responsible for 16:0-sphingolipid production. During brain development, one observes a 35-fold increase in the expression of CerS1 and a downregulation of CerS5 and CerS6 compared to embryonic tissue (Sladitschek & Neveu, 2019) (**Fig. S8A**). In contrast, during *in vitro* neuronal differentiation, between day 8 and 12, CerS1 expression increases only by 5-fold and, contrary to expectation, CerS6 expression is upregulated and CerS5 expression is unchanged (Gehre et al., 2020) (**Fig. S8B**). This could underpin the observation that 16:0-sphingolipids remain elevated whereas brain-specific 18:0-sphingolipids only increase marginally, despite supplementation with stearic acid. Overall, this suggests that appropriate programming of the sphingolipid metabolic machinery is not fully established in stem cell-derived neurons.”

Figure S8. Expression of ceramide synthases involved in sphingolipid biosynthesis. A) Expression levels are compared between E7 mouse embryos and pooled brain samples from P56-P84 mice. The data is from n =1 replicate and taken from RNA-seq (Sladitschek & Neveu, 2019). **B)** Expression levels are compared across the *in vitro* differentiation of mESCs into neurons in embryoid bodies. The data is from n = 5 replicates and taken from RNA-seq (Gehre et al., 2020). CerS1 and CerS4 are responsible for incorporating C18 chains into sphingolipids whereas CerS5 and CerS6 incorporate C16 chains (Levy & Futerman, 2010).

Minor comments:

1. I find the data presentation of the LENA analysis to be difficult to follow (**Fig. 1E**). In my opinion, the p-value is not the most important bit of information in this graph, though having it on the y-axis with other pertinent information encoded by colors or arrows being disguised. I would rather see the data on the x-axis that is above a certain p-value (denoted in the figure legend) plotted with the direction and magnitude of change shown.

We thank the reviewer for this suggestion. In the revised manuscript, we now plot $\log_2(\text{odds ratio})$ on the y-axis instead of the p-value. Moreover, we have dimensioned the size and color intensity of each point as function of the p-value (**Fig. 1E and 2E**, shown below).

Figure 1E. Lipid feature ENRichment Analysis (LENA) of features among decreasing and increasing lipids, where $-\log_2(\text{odds ratio})$ is plotted in the y-axis and the color and size correspond to the $-\log_{10}(\text{p-value})$.

Figure 2E. Lipid feature ENRichment Analysis (LENA) of the pool of decreasing and increasing lipids.

2. In the PCA in **Fig 1**, what are the loadings that define the variable PC1 and PC2? What is predominantly changing the P21 samples that lead to such a large shift if most of the data shown in the subsequent panels are not changing much between P2 and P21.

In the revised manuscript, we now include a plot of the PCA loadings of the lipids majorly influencing principal components 1 and 2 as supplemental **Fig. S3**, included here. Principal component 1 has a large contribution from cholesterol, which increases 1.5-fold between P2 and

P21 tissues. This could account for the large shift between P2 and P21 samples seen in the PCA plot.

Figure S3. Principal component loadings of lipids in the analysis of mouse tissues. PC1 and PC2 loadings of lipids where either loading was greater than 0.1 in magnitude are shown in this plot.

Reviewer #1 (Significance (Required)):

This work provides a nice reference for the complex lipidomes in embryonic and postnatal murine brain development. The details of the lipotype changes during development are well laid out and will of no doubt be of great use across a variety of scientific fields. While I found the in vivo data to be compelling, interesting, and useful, the lack of controls for the in vitro stem cell differentiation

work makes this particular data set and comparison less useful. Further work to identify the limitations of the stem cell differentiation protocols as a valid comparison to in vivo brain development need to be done and/or the discussion of the direct comparisons between the two toned down.

Reviewer #2 (Evidence, reproducibility and clarity (Required)):

The study used a quantitative lipidomics approach which I am very familiar with. The results should be highly reproducible.

Reviewer #2 (Significance (Required)):

The manuscript submitted by Gopalan et al. reported a quantitative and comparative lipidomics study between mouse brain samples from early embryonic to postnatal stages, and rodent and human stem cell-derived neurons. The authors found a couple of very unique characters only existing in brain samples, but not in stem cell-derived neurons, including 22:6-containing glycerophospholipids and 18:0-containing sphingolipids. The authors further found the brain-like lipotypes can only be partially established in stem cell-derived neurons after supplementing brain lipid precursors. These findings clearly suggest that stem cell-derived neurons might not be appropriately used to mechanistically study lipid biochemistry, membrane biology, and biophysics in brains. The study was well designed. and the manuscript was very informative and resourceful. I would suggest to accept the manuscript for publication.

We thank the Reviewer for the positive assessment of our work.

References

- Gehre, M., Bunina, D., Sidoli, S., Lübke, M. J., Diaz, N., Trovato, M., Garcia, B. A., Zaugg, J. B., & Noh, K. M. (2020). Lysine 4 of histone H3.3 is required for embryonic stem cell differentiation, histone enrichment at regulatory regions and transcription accuracy. *Nature Genetics*, 52(3), 273–282. <https://doi.org/10.1038/s41588-020-0586-5>
- Levy, M., & Futerman, A. H. (2010). Mammalian ceramide synthases. *IUBMB Life*, 62(5), 347–356. <https://doi.org/10.1002/iub.319>
- Sladitschek, H. L., & Neveu, P. A. (2019). A gene regulatory network controls the balance between mesendoderm and ectoderm at pluripotency exit. *Molecular Systems Biology*, 15(12), 1–13. <https://doi.org/10.15252/msb.20199043>
- Venkataraman, K., Riebeling, C., Bodennec, J., Riezman, H., Allegood, J. C., Cameron Sullards, M., Merrill, A. H., & Futerman, A. H. (2002). Upstream of growth and differentiation factor 1 (uog1), a mammalian homolog of the yeast longevity assurance gene 1 (LAG1), regulates N-stearoyl-sphinganine (C18-(dihydro)ceramide) synthesis in a fumonisin B1-independent manner in mammalian cells. *Journal of Biological Chemistry*, 277(38), 35642–35649. <https://doi.org/10.1074/jbc.M205211200>

January 29, 2024

RE: Life Science Alliance Manuscript #LSA-2024-02622

Dr Christer S. Ejsing
University of Southern Denmark
Biochemistry and Molecular Biology
Campusvej 55
Odense 5230
Denmark

Dear Dr. Ejsing,

Thank you for submitting your revised manuscript entitled "Lipotype acquisition during neural development in vivo is not recapitulated in commonly used in vitro neuronal differentiation protocols". We would be happy to publish your paper in Life Science Alliance pending final revisions necessary to meet our formatting guidelines.

- please be sure that the authorship listing and order is correct
- please upload your main manuscript text as an editable doc file
- please upload your main and supplementary figures as single files
- please add your main, supplementary figure, and table legends to the main manuscript text after the references section. Captions should not be included with the figures. Please separate the Figure legends ('Figure legends') and Supplemental Figure legends ("Supplementary figure legends") into separate sections
- Please add a Running Title in our system
- Please indicate secondary corresponding author on the submission page
- Please add a Summary Blurb/Alternate Abstract in our system
- Please add a Category for your manuscript in our system
- Please add the Twitter handle of your host institute/organization as well as your own or/and one of the authors in our system
- Please move the Table S1 at the end of manuscript file
- Please use the [10 author names, et al.] format in your references (i.e. limit the author names to the first 10)
- Please add a callout for each section of each Figure to your main manuscript text. (missing callouts: Figure 1B, Figure S1C,D,G, Figure S2 A-E, Figure S3 A-C, Figure S7A,C,D, Figure S8 C-E, Figure S9 C-E)
- Please add an Author Contributions section to your main manuscript text and please indicate the contributions in the system as well
- the Significance statement should be removed

A. FINAL FILES:

- An editable version of the final text (.DOC or .DOCX) is needed for copyediting (no PDFs).
- High-resolution figure, supplementary figure and video files uploaded as individual files: See our detailed guidelines for preparing your production-ready images, <https://www.life-science-alliance.org/authors>
- Summary blurb (enter in submission system): A short text summarizing in a single sentence the study (max. 200 characters)

including spaces). This text is used in conjunction with the titles of papers, hence should be informative and complementary to the title. It should describe the context and significance of the findings for a general readership; it should be written in the present tense and refer to the work in the third person. Author names should not be mentioned.

B. MANUSCRIPT ORGANIZATION AND FORMATTING:

Sincerely,

February 8, 2024

RE: Life Science Alliance Manuscript #LSA-2024-02622R

Dr. Christer S. Ejsing
European Molecular Biology Laboratory
Biochemistry and Molecular Biology
Campusvej 55
Odense 5230
Denmark

Dear Dr. Ejsing,

Thank you for submitting your Research Article entitled "Lipotype acquisition during neural development is not recapitulated in stem cell-derived neurons". It is a pleasure to let you know that your manuscript is now accepted for publication in Life Science Alliance. Congratulations on this interesting work.

DISTRIBUTION OF MATERIALS:

Again, congratulations on a very nice paper. I hope you found the review process to be constructive and are pleased with how the manuscript was handled editorially. We look forward to future exciting submissions from your lab.

Sincerely,
